# Prospects for Electric Vehicles

**Jack N. Barkenbus**

Vanderbilt Institute for Energy & Environment, Vanderbilt University, Nashville, TN 37240, USA;
jack.barkenbus@vanderbilt.edu

**Abstract:** The transformation from conventional vehicles to electric vehicles is underway, but its timeline is still uncertain. The battle against climate change provides the essential backdrop, leading governments to encourage this transformation by providing both consumer incentives to purchase electric vehicles and by establishing regulations requiring automakers to produce them. Behind this prodding are a set of fundamental forces that both encourage and discourage consumer interest, as set forth in the text. Fortunately, intensive battery research and development is proceeding that should alter market forces and make electric vehicles more attractive to segments of the population not now in the market for an electric vehicle. Hence, even if the timeline for battery improvement cannot be predicted with certainty, continued government support, and upstart automakers, such as Tesla, should ensure that the transformation will proceed over time.

**Keywords:** electric vehicles; sales; diffusion; advantages; shortcomings; incentives; regulations

## 1. Introduction

The transition to electric vehicles (EVs) is underway. While the transformation from conventional passenger vehicles (those with internal combustion engines, powered by oil-based fuels) to EVs (4-wheel plug-in vehicles propelled totally, or in part, by electricity [1]), is certain, the trajectory of this transition is far from settled.

EVs have a long history, and were quite prominent at the beginning of the automotive era [2]. The technology succumbed, however, to the business model set in motion by Henry Ford for the conventional passenger vehicle, and has lingered in the shadows ever since. Reasons for its dormancy are not only technical but the fact that it never, until now, engaged a supporting socially-constructed worldview that would create a powerful incentive for its ascendency [3].

The difference now is that both profit-making firms and countries perceive a commercial interest in EVs, and, most critically, EVs are perceived by larger society as a key force for combating climate change. Decarbonization strategies typically include the transition to electric transportation as a key element in long-term plans [4,5].

Specifically, the spark for a new transportation era has been provided by two entities: the audacious and improbable rise of the EV maker, Tesla [6], and the ascendency of China, and its bevy of EV startups, as the global EV hotspot. Over half of all EVs sold in 2018 and 2019, were sold in China, which perceives a new and enormous commercial opportunity arising from the transition [7].

Though understandably slower out of the gate, conventional automakers have now announced ambitious plans for electrification of their respective fleets, some of which are outlined in Table 1.

**Table 1.** Key Automaker Announcements for Electric Vehicles *.

| Automakers | Announcements |
| --- | --- |
| BMW | Plans to offer 12 EVs by 2025 |
| Ford | Plans to offer 40 EVs by 2022 |
| General Motors | Plans to offer 20 EVs by 2023 |
| Mercedes | EV sales are estimated to comprise 15–25% of total sales by 2025 |
| Volkswagen | Plans to offer 80 EVs by 2025 |
| Renault-Nissan | EV sales are estimated to comprise 20% of total sales by 2020 |

* Adapted from Electric Power Research Institute, Consumer Guide to Electric Vehicles, March 2019.

Dozens of EVs have already reached the market and over seven million are on roads across the globe today [8]. Yet sales are still exceedingly small relative to the overall market for passenger vehicles. In 2019 [9], sales of EVs totaled 2.2 million, just a 2.5% share of the market—meaning that only 1 in 40 passenger vehicles sold was an EV. Some countries, such as Norway, Iceland, the Netherlands, and Sweden, sold more than 10% EVs in their respective markets [8] (Norway even had sales exceeding 50%). Yet the relatively small populations in these countries means that these sales hardly impacted global automotive markets in any meaningful way. It is safe to say, therefore, that the transition to EVs is in a very early stage.

One way to think about this state of affairs is through the standard Rogers product diffusion model [10], as shown in Figure 1.

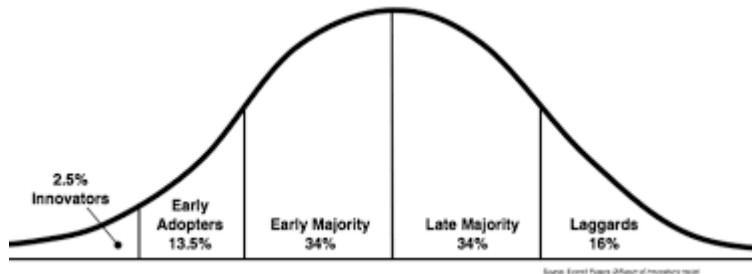

**Figure 1.** Product Diffusion Model.

With sales of 2–3 percent, EVs are still in the beginning stage of diffusion, attracting only the "innovator" or the beginning stages of the "early adopter" segments within a given population. Individuals within these segments of the population differ from those within the more general population. They are open to experiencing new technologies, are willing to overlook imperfections in the technology in order to achieve personal or societal goals, and are not swayed by mass marketing. Numerous projections for when (or if) EVs will move up the diffusion curve have been attempted. A recent study that compared these projections in terms of the percentage of EVs sold twenty years from now (2040), revealed the range of these projections were from 10 percent to 70 percent of total market share [11]. This clearly shows that there is no consensus or conventional insight into how swiftly EVs will come to dominate the passenger car market.

The absence of consensus is a reflection of the contradictory forces now at play. While the author believes a quick transition to EVs is paramount for dealing with the major public policy issue of our time, climate change, there are no guarantees that a quick pace will come to pass. The pace is a product both of technological progress and societal reckoning with the challenge of climate change. This article reveals the major factors at play in determining the pace of the transition. Subsequent sections will outline why EV adoption rates are meager to date and the forces at work to both advance sales and retard them. At the

current time, strong governmental support for EVs is propelling sales, but ultimately the pace of this transformation will depend upon more favorable market forces.

## 2. Societal Impact

The underlying case for EVs rests on the fact that $CO_2$ emissions from the transportation sector are large and growing faster than emissions from other sectors. Emissions are directly attributable to the carbon-emitting fuels powering the sector, namely, the oil-based products gasoline and diesel. In 2018, the transportation sector was responsible for approximately 14 percent of all Greenhouse Gas Emissions (GhGs) and a quarter of the emissions derived from burning fossil fuels [12]. Passenger vehicles are responsible for a large part (72%) of the sector's emissions and are the primary reason for the sector's rising emissions [12]. Fuel substitution, therefore, is a critical element in reducing the climate impact emanating from the transportation sector. The impact of fuel substitution can, of course, be complemented by increasing fuel efficiency, car sharing, public transportation, etc. However, the ubiquity of personal ownership and operation of automobiles is all-pervasive (some have claimed we have created an "auto-centric" society based on auto-mobility) [13] and seems unlikely to diminish any time soon. The number of automobiles on our roads reached 1 billion in 2010 and now stands at 1.25 billion. Some projections now foresee a total of 2 billion autos on our roads by 2035 [14].

Second, it is now well established that EVs make a considerably smaller $CO_2$ impact, across the vast majority of countries, than conventional passenger vehicles [9,15,16]. How much cleaner EVs are is a complicated calculation involving several variables, including EV and conventional vehicle operating efficiencies, and carbon levels of the electricity being delivered to EVs (as well as carbon levels of refined oil products). The mix of primary fuels in the production of electricity varies from country to country and city to city within countries. The International Energy Agency (IEA) has calculated that the current average carbon intensity of global electricity is 518 $gCO_2$/kWh, a quantity at which typical EVs are cleaner than most conventional vehicles [9]. Again, this average does not account for the large variation in carbon-intensive electricity across countries, such as the variation between Poland and Norway. The good news is that with the expected greater use of renewable energy in the production of electricity in the future, EVs will become even cleaner over time. Developments in the power sector, therefore, will have a direct bearing on the societal benefits we can expect from the EV transformation within the transportation sector.

As the electric grid becomes cleaner, EVs will also make a significant contribution to reducing local air pollution, which remains the deadliest of all impacts resulting from the burning of fossil fuels [17]. Once again, the calculation of the EV impact on reducing local pollution is complicated. The absence of tailpipe pollutants in vehicles traversing dense, urban populations is an obvious benefit. Yet, it if means the production of more dirty electricity at the power plant, and if the plant's emissions drift into these dense urban conclaves, little positive air pollution results will be achieved. Site-specific analysis, therefore, is necessary to determine the extent of air pollution benefits derived from the EV transformation. Perhaps the best way of thinking about global and local air pollution impacts of the EV rollout, therefore, is to recognize that its significance is dependent upon synergistic and interconnected improvements in electricity generation.

Still another benefit of the EV transformation will be the reduction in oil imports resulting from diminishing gasoline and diesel sales [12]. Large-scale oil production is centered in relatively few countries, some of whom are politically unstable. This instability creates conditions leading to price fluctuations and supply insecurities. Electricity, in contrast, can often be produced from domestic resources, particularly if these resources are renewable. Backing oil out of the energy supply system, therefore, can lead to greater price and supply security.



In summary, there are good reasons to believe that the movement to EVs will result in significant societal benefits. Having said that, however, it is important to remember that unlike "innovator" or "early adopter" segments of the population, the broader public will generally make automobile purchases not on the basis of achieving societal goals but rather according to personal taste. Owners want to believe that there are specific advantages that accrue to them alone through the purchase of one vehicle model as opposed to another. These perceived advantages may be functional (such as affordability, reliability, and comfort) or may be symbolic (indicative of social status or group membership) [18].

It is important to point out, therefore, that there are already some functional and symbolic advantages of EVs that will attract the general population. In the section that follows, the personal benefits described pertain to owners of fully-electric vehicles (BEVs). Those owning hybrid plug-in vehicles (PHEVs) will, of course, see fewer benefits from those described.

## 3. Personal Benefits

Even consumers least concerned about the environmental impacts of their vehicle's operation can find personal advantages to owning an EV. The first is that EVs are cheaper to power than conventional vehicles [19,20]. Electricity and oil prices vary across the globe, but generally, travel in an EV is less expensive than in an oil-based vehicle. In the United States, a trip of 200 miles will cost the EV traveler $7.42 (using average electricity fares and average EV vehicle efficiency). Using a conventional vehicle, the cost will be $22.60 (again, using average fuel prices and average fuel efficiency). In addition, consumers can count on electricity prices being more stable over time, unlike the volatile oil market [21].

Second, maintenance costs, and the inconveniences associated with servicing the conventional vehicle, are considerably fewer with an EV [9,19,20]. Conventional vehicles have thousands of moving parts, all of which have finite operating lives and need to be synchronized for optimal performance. EVs, in contrast, have few, perhaps two dozen, moving parts, and thus there is less opportunity for vehicle malfunction. EVs, for example, contain no lubricating oils, filters, clutches, spark plugs, pistons, timing and fan belts, water hoses, radiators, or catalytic converters.

Third, EV owners with owned or rented houses have a built-in infrastructure for powering their vehicles; namely, the house's electrical system. The EV is no different than other common household electronics in that it can be charged overnight and be fully ready to operate in the morning. For urban driving, therefore, the EV owner never has to utilize external, commercial charging stations. Long-distance driving will require highway stops, as with conventional vehicles. While highway charging outlets are currently not as ubiquitous as gasoline or diesel refueling stations, industry and government sources are active in building out the infrastructure that will eventually match and replace the convenience of fueling stations [22,23].

Finally, EVs offer a better driving experience. To some, this means faster acceleration. Tesla's brand has been enhanced considerably by its vehicles' ability to accelerate from 0 to 60 mph in an incredible 2–3 s. While other EVs cannot quite match that performance, all BEVs demonstrate quick acceleration, a consequence of the greater propulsion efficiency of the EV relative to conventional vehicles. To others, a better driving experience may be manifest in a quiet ride, a consequence of eliminating the combustion and moving parts associated with conventional vehicles [24,25].

Unfortunately, the foregoing advantages to EVs just described are not the end of the story. There is a countervailing set of facts that make the purchase of an EV less attractive. We might characterize them as barriers to adoption, both intrinsic to the vehicles themselves and extrinsic to the vehicles.

## 4. Intrinsic Shortcomings

While EVs have advantages, they do not currently offset or substitute for all the desirable features we have come to expect from conventional vehicles. These deficiencies will have to be overcome if EVs are to appeal to the "early majority" segment of the buying population, and even more so to the subsequent segments. The fundamental problem relates to the state of battery development. Power-delivery batteries are still relatively expensive to produce, requiring automakers to charge a premium for BEV purchase. How large a premium can vary from model to model, but it inevitably increases the first-time costs of EVs. It may be that the lifetime costs of a EV are less than those of a conventional vehicle, due to its lower operating costs, but surveys have shown that the initial purchase price of a vehicle plays an inordinate role in shaping consumer purchase decisions [26]. Short-term perspectives, therefore, trump long-term considerations. No surprise there.

Batteries currently have other downsides, such as the inability to provide comparable driving range to that of conventional vehicles [27,28] The 400+ mile range of conventional vehicles between fuel fill-ups cannot currently be matched by BEVs. The energy density associated with EV batteries is simply insufficient to attain such a range in distance. It may be, as frequently noted, that the vast majority of daily automotive trips can easily be met with current EV ranges. However, most drivers want the flexibility that comes with the 400+ mile range, to accommodate unanticipated contingencies in daily operation and to facilitate the occasional long-distance trip.

A key factor in reducing the resulting "range anxiety" of EVs besides through battery improvements is the placement of easy-to-access and ubiquitous public charging stations. While the pace of charging station installations has been increasing, its pervasiveness still comes nowhere near matching the number of fueling stations [29]. Further, there needs to be greater uniformity in stations that are being deployed, such that drivers need not worry whether their individual vehicle can re-power from the station [30]. A standard interface would help, but drivers also need high-speed charging. No one wants to spend an hour or more at the charging station on long-distant trips. Until the charging experience with EVs can match that of our current fuel-filling vehicles, the appeal of EVs will remain limited [31].

## 5. The Exterior Barriers

The forgoing perceived deficiencies result from shortcomings with EVs relative to conventional vehicles. However, there are other barriers to acceptance derived from forces extrinsic to the vehicle itself. While these forces are unlikely to derail the EV transformation, they could delay it, and, as such, deserve some attention.

The first barrier worth mentioning is the opposition from the automotive industry itself. While much of the industry has projected EV models being produced in the near future, as noted previously in Table 1, delay is still very much possible because there is no compelling economic advantage for the industry to switch to EVs. In fact, quite the opposite. It is well known that the profit margins from the sale of the most popular vehicles, i.e., large, roomy conventional vehicles, are substantial [32,33]. The global coronavirus pandemic is likely to encourage beleaguered automakers to bolster sales of its high-profit conventional vehicles at the expense of a swift turn to EVs.

The emergence of EVs, therefore, is a disruptive force, championed by upstart EV companies like Tesla, which has no conventional vehicles in its fleet to compete with its EVs. The unique contribution of Tesla is its demonstration effect; that is, the ability to produce an EV that consumers value and will purchase. Of course Tesla has yet to produce a vehicle for all strata of society, but it has popularized the notion of an EV future, and, as such, has disrupted the business model of the automotive industry [34].

Industry announcements of the scores of EVs now theoretically "on the market" are misleading at best [35,36]. Far fewer are typically found available in most locations, as the industry has no desire to sell

EVs in other than the major markets or where regulations compel them to do so. The relative absence of promotional marketing is a manifestation of this disinterest [37].

The segment of the automotive industry most opposed to EVs is the dealership. Tesla again has demonstrated that it can operate successfully without a traditional dealership network. Teslas are sold either online or through "showrooms" and, as such, constitute an existential threat to conventional dealers. Were EVs sold through a conventional dealership model, they would still represent an existential threat to dealers, since the dealer's business model is predicated on lucrative post-sale services, namely repairing and maintaining the vehicle. Since EVs have a dramatically lower maintenance profile, they significantly reduce dealer profits associated with post-sale services. It is no surprise, therefore, that stories of consumers entering a dealership asking to be shown an EV and subsequently directed to conventional vehicles are legendary and widespread [38].

Another extrinsic force seeking to eliminate the emergence of EVs is the fossil fuel industry. Again, it results from the fact that EVs pose an existential threat to the industry by displacing the oil-based fuels, gasoline and diesel. Since approximately three-quarters of a barrel of oil is refined as either gasoline or diesel, the loss of volumes now directed to the automotive sector would be an enormous setback for the industry. Funding of studies purporting to show the folly of any EV transformation, by the fossil fuel industry, has been substantial [39].

Another force holding back EVs is not an industry, but a mindset. A certain fraction of the population has come to revere conventional vehicles based on their "romance" with internal combustion engines. To these people, automobiles and internal combustion engines are inseparable. EVs introduce a new and foreign world without the familiar trappings of their youth. There is some overlap between these people and those engaged in a cultural war over climate change. Some believe that the movement against climate change is simply a ploy seeking to overturn established society and impose a new world order. Since EVs are seen as playing a central role in the climate change story, they are consequently perceived as a "trojan horse" aimed at disrupting our existing way of life [40]. Those who hold this point of view are not really interested in discerning the technical issues associated with the pros and cons of EVs, but rather are inclined to seek out arguments that will discredit the transformation altogether.

As noted previously, these extrinsic forces can delay the transformation but, by themselves, are not strong enough to derail it, should the current intrinsic shortcomings of EVs be overcome. Further, as we will see in the next section, there is certainly good reason to believe that they can be overcome.

## 6. The Outlook

Despite the factors mitigating against EVs, there are grounds for optimism when assessing the future. Tesla has made it known that it will continue on the EV path regardless. There are numerous budding Chinese automakers who will be willing to take Tesla on in the future. Further, Volkswagen has already announced its intention to transition to electric car manufacturing after its next round of conventional vehicles is complete [41].

The factors compelling automakers to herald electrification are both technical and political. On the technical side, there is the expectation that EV batteries will decrease in cost while simultaneously expanding driving range. This expectation is built on experience to date. In 2010, EV battery packs cost over \$1000/kwh to produce; and yet by 2019 (as shown in Figure 2), costs had declined to approximately \$200/kWh [42]. Conventional wisdom now has battery pack costs falling to approximately \$100/kwh at some point in the 2020–2030 time range. Figure 2 is a widely-publicized forecast made by BloombergNEF in 2019—based on surveys of dozens of industry stakeholders—that predicts battery pack costs will go below \$100/kwh as early as 2024 [43]. This is the cost at which EVs are expected to be competitive with

conventional vehicles on a first-cost basis, thereby significantly altering the consumer calculus to purchase an EV. General Motors [44] and Volkswagen [45] projections are similar.

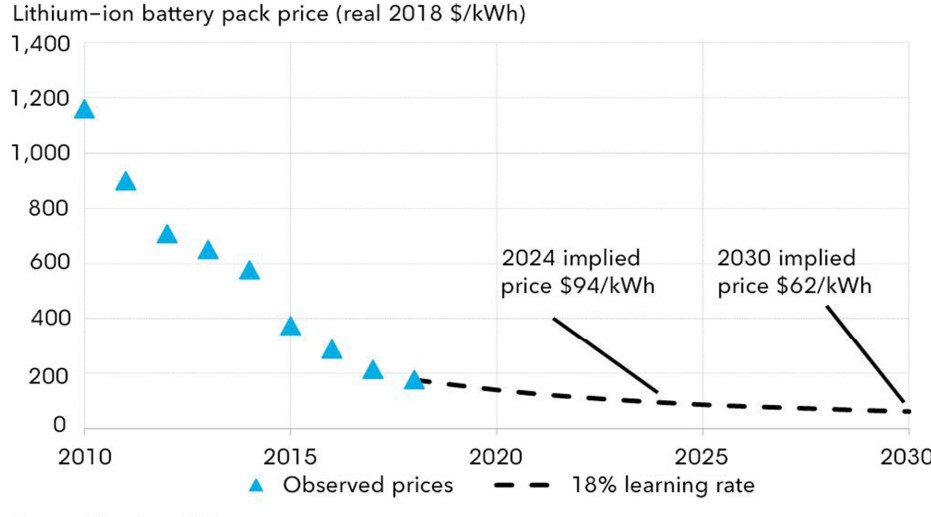

**Figure 2.** Lithium-ion battery pack outlook.

Not everyone agrees. Tesla, for example, feels that this key price point can be reached sooner [46]. On the other hand, a peer-reviewed article claims that it will be reached during the second half of the decade [42]. Further, an MIT study claims that it will be at least 2030 before the $100/kwh benchmark can be reached [47].

Not everyone believes the lithium-ion battery—that has dominated EV power installation to date—is capable of achieving this price point while simultaneously increasing the vehicle's range; but research on alternative battery configurations is intense (both in academia and industry) and these alternative battery chemistries should be able to succeed over time should lithium-ion batteries falter [48–50]. EVs produced a decade ago typically had a driving range of 100 miles or less between charges. Today's EVs, with bigger and more powerful battery packs, have expanded driving range past 200 miles and greater distances are anticipated in the future. Tesla already has a model said to reach a 390 mile range [51]. Optimism regarding battery development emanates not only from the automobile sector but from the electric power industry's concurrent efforts to make storage a valuable new component of the energy system [52].

## 7. Incentives

In light of current battery shortcomings, governments across the globe have attempted to offset these deficiencies with incentives for the purchase of EVs. Without these incentives, EV purchases would have been even more limited than what we have witnessed to date. Future battery improvements may make these incentives less necessary but, for the time being, there is little indication that government incentives are going away. Indeed, societal benefits from the EV transformation are so compelling that incentives appear to be a stable feature of the automotive environment [53].

Governments vary in their aggressiveness with respect to incentives. Listings can be found at [9,53,54]. At the lower end of the spectrum are measures intended just to build awareness of EVs within the general public. This is a low-cost and somewhat useful strategy, since there is still a great deal of public ignorance of EVs. Manifestations include the construction of a website where all things EV are set forth or the

proclamation of "EV Days", where the public can come to a central location and view many of the EV models currently in the market. It might even allow the public to test drive an EV, thereby allowing an interested public the opportunity to see what it feels to drive an EV. Awareness can also be built through government purchases of EV motor fleets for employee work-related tasks. As useful as the awareness strategy is in laying the foundation for EV sales, it is hardly sufficient in itself to make a significant difference in sales volumes. Consumers need a more compelling value proposition than can be achieved just through better education.

Moving further along the aggressiveness spectrum, governments can assist in building a better infrastructure conducive to normalizing EV operation. Most prominently, it can build more charging stations, thereby reducing the "range anxiety" associated with driving vehicles with limited range. As mentioned previously, charging stations are being built in all countries, but they still do not match the number of fueling stations currently available. More needs to be done. Other actions governments are taking to make EV operations more convenient include distributing special parking permits available only to EV owners, and in urban areas, they can make EVs eligible for travel in specially-designated High Occupancy Vehicle (HOV) lanes.

One-time purchase subsidies for EVs constitute an even more aggressive government strategy. These may come in the form of instant rebates or the more delayed income tax credit. Since the purchase price of an EV is a primary obstacle to greater EV sales, this incentive is particularly effective in nudging EV sales upward. Other financial incentives often applied to EVs include tax breaks on annual EV registrations and special EV electricity charging rates targeted to EV owners. While financial incentives can be one of the most important actions governments can take to increase sales, opposition can form, arguing that tax monies should not be used to support purchases by the well to do [55,56].

Perhaps the strongest message governments can opt for is to actually impose disincentives on the purchase and operation of conventional vehicles. Disincentives have been imposed by several nations with the highest percentage of EV sales, such as Norway, China, and France. In Norway, imported conventional vehicles are subject to a stiff import tax, while EVs are not [57,58]. Several Chinese cities have imposed driving bans on conventional vehicles during certain days of the week, while EVs are free to roam. Moreover, several of these cities have imposed hefty fees on the acquisition of driver licenses for conventional vehicles while waiving them on EVs [59,60]. Further, France has imposed fees on vehicles calibrated to their $CO_2$ emissions. Vehicles with higher emissions are saddled with higher fees [61]. Obviously, a public generally attuned to the dangers of climate change is a prerequisite for installing meaningful disincentives on the purchase of conventional vehicles.

While incentives can nudge the public in the direction of greater EV sales, they are not, even at the most aggressive end, the most consequential action governments can take to increase EV sales. For that, we turn to regulatory actions.

## 8. Regulations

Incentives are designed to change the consumer's value proposition. Regulations, in contrast, require suppliers to produce EVs irrespective of consumer sentiment. The theory is that once manufacturers are required to produce EVs, they will have a vested interest in promoting them. Manufacturers, of course, dislike being told what to produce, and will argue that they cannot sell what the public is not demanding; such arguments can be ignored if there is sufficient political will.

At the current time, that political will exists within three key government jurisdictions: the state of California, China, and the European Union. Collectively, these jurisdictions cover a little more than half of the automotive sales market and, as such, manufacturers simply cannot ignore or dismiss regulations imposed.

### 8.1. California: Zero Emission Vehicles

California has promulgated an aggressive incentive program for EVs which has, in part, resulted in the state capturing approximately half of all EV sales in the United States. Approximately 7 to 8 percent of all passenger vehicle purchases in California are EVs. However, the state long ago recognized that a regulatory mandate was required in addition to consumer incentives [62]. Toward that end, it created the Zero Emission Vehicle (ZEV) regulation in 1990, requiring each automaker with sales in California to produce and market a certain percentage of vehicles having no tailpipe emissions. This was put into force to deal with California's serious air pollution problems even before climate change came to dominate the public agenda.

This credit-based regulatory system has gone through various incarnations over its thirty-year history and, because of its complexity and regulatory flexibility, has attracted its share of critics [63–65]. Nonetheless, it has resulted in a much larger number of EVs being marketed in California, as opposed to other states, and creates a degree of certainty amongst manufacturers that even larger numbers will be required in the future. Ten other states have adopted California's credit-based system, ensuring greater market penetration for EVs even in the absence of a national ZEV standard [66]. Purchases in ZEV states collectively add up to nearly one-third of the U.S. automotive market.

### 8.2. China: New Electric Vehicles

In 2017, China launched its New Electric Vehicle (NEV) regulatory program based largely on California's credit-based ZEV program. It has set a goal of EVs constituting 12 percent of all vehicle sales in 2020, moving up to 14 percent in 2021 [67,68]. Looking even further ahead to 2025, the goal is 25 percent of all sales. It must be remembered that these goals are expressed in volumes, while the system is based on credits earned. China is enthusiastic about EVs for several reasons, including the opportunity to compete successfully against foreign brands. Further, just as in California, an impressive consumer-based incentive system complements the regulatory NEV.

Since China is the largest market for passenger vehicles in the world, foreign automakers do not wish to cede the lucrative market to Chinese EV manufacturers. Much of the impetus for EV manufacturing globally, therefore, can be traced to the desire to remain competitive in China.

### 8.3. European Union: Passenger Vehicle Emission Standards

Europe has taken an alternative approach to fostering EV growth from the U.S. and China [69]. Europe's approach is still credit based, but rather than setting numerical goals for EV volumes, the European Union has set numerical $CO_2$ emission standards with the expectation that these standards can only be met by greater EV market penetration. The EU standard for passenger vehicles is no more than 95 g$CO_2$/km, to be achieved by 2021.

The credit-based system is predicated on the requirement that each automaker will produce a fleet average of 95 g. Excess emissions from some conventional vehicles can be offset by the "supercredits" that automakers can earn by the sale of EVs. The inability or unwillingness of some automakers to meet the emissions target can also be compensated for by buying credits from those automakers who have "overperformed." Emission standards are set to become stricter in the future, as the 2025 target has been set at 80 g/km.

### 8.4. Government Phase Out or Bans of Conventional Vehicles

For critics of the flexible, credit-based regulatory systems just described, there is another, and seemingly firmer, measure being promulgated by some governmental jurisdictions: namely, an outright ban or projected phase out of conventional vehicle sales at some time in the future. At the current time, 14 countries

and 20 cities globally have produced declarations of intent that would trigger these bans [70–72]. Typically, the timetable for such bans is in the 2030 to 2040 time range. Denmark and France, for example, have set 2030 and 2040 timelines, respectively. London and Los Angeles have chosen 2030. Regardless of details, these bans send a powerful political message that business as usual is set to change in the future.

## 9. Conclusions

In summary, major governments are not passively waiting for an EV transformation, but are actively promoting it through consumer incentives and government regulations. There is a wave of regulatory actions in large automobile markets—fostered by greater concern being given to climate change—designed to scale the transformation as soon as possible. Vested interests may still seek to derail it, but, at most, all they can do is delay it.

Political forces, however, will still have to reckon with market forces. Should the conventional wisdom regarding future battery improvements be unwarranted, political winds will change. The thirty-year history of California's ZEV regulation is a case in point. When market forces did not line up with calendar deadlines, politicians took the route of flexibility by moving the deadlines further into the future. Regulations and bans, therefore, send an important signal to the marketplace, but should be viewed in an aspirational, rather than an absolute, sense.

So even if the transformation to EVs is well underway, we still do not know the timeline governing it. Political bans on conventional vehicles are meaningless as something more than a political signal. Industry timelines for the introduction of specific EV models are even more suspect, witnessed by the recent industry backtracking on the introduction of fully autonomous vehicles [73], and the likely fallout from the global coronavirus pandemic. The transition will probably occur earlier in some regions than others. China looks to be the first region making the transition because of its significant incentives and regulations, the absence of strong competing fossil fuel interests and conventional automakers, and what it perceives as a strong export potential in both batteries and automobiles. Europe will probably be next, again because of its incentives and regulations, as well as a heightened determination to combat climate change. North America (excepting California) is destined to be the laggard. Nonetheless, momentum for EVs will not fully dissipate as long as climate change looms as large in our consciousness as it does today.

**Funding:** This research received no external funding.

**Conflicts of Interest:** The author declares no conflict of interest

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
