# Peer review of "Prospects for Electric Vehicles"

_sustainability, doi:10.3390/su12145813_

Round 1

Reviewer 1 Report

This paper reviewed factors that have propelled or impeded the uptake of electric vehicles. This review has made a limited new contribution to the literature, as more detailed reviews have already been conducted by various studies, focusing on, for example, factors that influence consumer intentions to adopt electric vehicles.

Author Response

Thank you for reviewing the manuscript.  Yes, I agree with you that the article makes a "limited new contribution to the literature."  It is intended to be a review overview aimed not at experts in the field but others who may have an interest in the field but lack a background to evaluate the "big picture."  Those wanting a more detailed review can go to the footnotes for a start in discovering more detail.

Reviewer 2 Report

The Author at work analyzes the development of the electric vehicle market. The analysis includes information from other publications. The analysis took into account factors such as vehicle range, environmental and legal considerations that have an impact on the development of the market for this type of vehicle. The analysis applies only to passenger cars, at work there is no information about commercial vehicles such as buses, construction vehicles, heavy duty vehicles, etc. Since the publication is cross-sectional and does not bring any new knowledge, it is only a collection and summary of available knowledge about the electric vehicle market, it is justified to supplement it for information on the market of other electric vehicles, not only personal ones. After completing this information, the work may be an interesting popular scientific position describing the anticipated development of the electric vehicle market.

Author Response

Thank you for reviewing the manuscript.  Agreed, "it is only a collection and summary of available knowledge about the electric vehicle market."  Indeed, I think that is basically the purpose behind a review article.  As for going beyond the passenger-only market for electric vehicles, that is a paper for another day.  A worthy topic, no doubt, but a completely different manuscript.

Reviewer 3 Report

It is well—written and I agree with all evaluations and conclusions presented by the author. I recommend publication.

If as a reviewer, I have to make “critical” remarks, I would put forward these three:

  • It is not an academic paper. It presents a general point of view instead of focusing on a specific topic.
  • Line 109 “the general public will make automobile purchases not 109 on the basis of achieving societal goals but rather according to personal taste”. Yes, but individual have also social and political preferences (concerns for the future generations, ….). I think it should be stated.
  • It is rather US (Tesla)-centered. In other parts of the world, the forces at play might be different. In Europe, for instance, VW is playing and will play a leading role in pushing towards BEVs, even in convincing politicians and favoring technology-specific regulation. In China, it is a different story. The Chinese government promoted EVs to have a competitive advantage over foreign car manufacturers. And so on.

Final question to the author. What drives the development of battery technology? Market forces or government-supported research?

Author Response

Thank you for reviewing the manuscript.  Agreed, it is not an academic paper, per se.  As a review paper it is intended to summarize available knowledge and, as you say, have a general point of view.

I've qualified the wording on line 109 to reflect your concern over overstating the personal vs. societal distinction.

Yes, it is U.S. oriented, I'm afraid, although I've tried to bring an international perspective.  Please note that I've added to the conclusion a more global perspective as to where global regions stand with respect to the transition.  Also note that while Tesla is a U.S. based company, its impact goes well beyond the U.S. with its popularity in Europe and its sales and manufacturing plant in China.

As to what drives the development of battery technology, I believe the answer is clearly market forces.  Government research on batteries is useful but there are sufficient incentives for market forces to push the technology regardless.  Government incentives and regulation play a critical role (more than government-based research) in jump-starting the transition, but, as I say in the paper, ultimately EVs must meet the market test.

Reviewer 4 Report

This paper reviewed the benefits and barriers in developments of electric vehicle market. However, it is hard to conclude the core thoughts from the author. A few suggestions are shown below for the author to improve the paper quality.

  1. Please list the most important views from the authors in the abstract and conclusion. Though it is a review paper, it does not mean the author should not present its own thoughts.
  2. What is possible market penetration in 2020, 2030 in each region? As an article talking about the “prospects of electric vehicles”, it did not discuss any potential market in the future.
  3. More reviews on battery cost are very needed. The author presented the battery costs in Figure 2, however, the only source is from BloombergNEF which is very aggressive on the battery cost projection. So, what are other projections of battery cost?
  4. Section of Regulations. Can the author list a table to compare the commons and differences among the countries or regions in the electric vehicle market?

Author Response

Thank you for reviewing the manuscript.

I believe my core thoughts on the technology come through clearly.  First, we are indeed undergoing a transition to EVs, even though the timeline of this transition is still uncertain.  On line 55 I state that my preference is for a quick and early transition because of the dangers of climate change.  I am optimistic about battery development, but acknowledge that this optimism may be upended.  I have a folder in my office titled "Electric Car Graveyard" that chronicles previous efforts to bring EVs to the mainstream that have now gone bust. 

I've changed the conclusion to deal with possible market penetration rates in the most important regions, i.e. China, Europe, and North America.

I've added to the battery cost discussion.  The BloomberhNEF scenario is optimistic, but it has become conventional wisdom.  Outliers are those who say we will reach the $100/kwh benchmark earlier (primarily Tesla observers) and those who say it will be later (such as the MIT study now referenced in the paper).

A new table comparing regulatory commonalities and differences amongst countries would be a little overwhelming, I'm afraid, because of inherent complexities.  I now point the reader to such listings or tables from other, larger studies than mine.

Round 2

Reviewer 1 Report

It would be useful if some discussion could be added in the introduction sector about the rationale for an overview of various factors that have shaped the process of transport electrification. 

Author Response

Thank you for the suggestion.  I have added a couple of paragraphs in the Introduction which I believe are consistent with your request, and provide additional context to the reader.  Thanks

Reviewer 2 Report

I accept the arguments provided by the author that a paper including a commercial vehicle market analysis would be too extensive. I believe that although work does not bring any scientific knowledge, it is an interesting summary regarding the development of the electric vehicle market. Therefore, it is an interesting popular science character literary position. I look forward to the next publication regarding electric commercial vehicles.

Author Response

Yes, at some time I would like to provide a review of commercial vehicle prospects.  A major problem in doing so is the fact that there are at least four distinct commercial applications, each with its own characteristics and nuances.   There is the long-haul industry; the passenger fleet sector (both governmental and private, such as Uber and Lyft); the final-stage delivery service, and; the electric bus sector.  Whether I can combine all in one paper is something I will have to consider.

Reviewer 4 Report

I do not satisfy with the replies from the authors on the battery costs. I totally understand the future trend of battery cost is hard to project. However, it does not mean the authors can project the battery cost according to personal feelings - "The BloomberhNEF scenario is optimistic, but it has become conventional wisdom." without any reference support. Most of BloombergNEF reports are not peer reviewed.

To improve the paper quality, I strongly suggest the authors to review more battery cost papers and use the conclusions in these paper to draw a figure on battery cost trend, instead of using the figure from BloombergNEF.

Besides, the formats in the manuscript need revision.

Author Response

I've reviewed the section on battery costs and now claim that conventional wisdom on the $100/kwh benchmark covers the entire 2020-2030 decade; more in line with your concern and, indeed, based on some extended reading.  I've retained the BNEF figure.  It is based on the firm's extensive network of sources and not just a personal assessment.  I recognize, however, it is still somewhat optimistic.  Personally, I think the barrier will be breached even sooner than Bloomberg, for some battery and automakers, but this is just a personal assessment based not on any inside knowledge but a perception that technical progress is moving faster than is generally perceived.

Round 3

Reviewer 4 Report

The authors have addressed the concerns I have, and edited the context in the manuscript.